# COVID-19 Fear, Resilience, Social Support, Anxiety, and Suicide among College Students in Spain

**DOI:** 10.3390/ijerph18158156

**Published:** 2021-08-01

**Authors:** Jesús Muyor-Rodríguez, Francisco Caravaca-Sánchez, Juan Sebastián Fernández-Prados

**Affiliations:** 1Department of Psychology, CEMyRI, Area of Social Work and Social Services, Almería University, 04120 Almería, Spain; jesusmuyor@ual.es; 2Department of Psychology, Area of Social Work and Social Services, Almería University, 04120 Almería, Spain; 3Department of Sociology, CEMyRI, Almería University, 04120 Almería, Spain; jsprados@ual.es

**Keywords:** fear, COVID-19, resilience, social support, anxiety, suicide risk, college students, Spain

## Abstract

Worldwide, previous studies have expressed concerns regarding the broad psychological effects of the COVID-19 pandemic among college students as they are considered an especially vulnerable group. However, few studies have examined the prevalence of, risk of, and protective factors associated with fear of COVID-19 among college students in Europe. We have sought to address gaps in the literature by conducting a cross-sectional survey among 517 college students (79.1% women and 20.9% men) from a public university in the southeast of Spain. Participants were asked to complete the Fear of COVID-19 scale (FCV-19S) questionnaire and answer questions related to resilience, social support, anxiety, and suicide risk levels using validated scales. The results of the analysis of the variables associated with fear of COVID-19 suggest that, in addition to gender, the factor of anxiety shows a robust positive association and effect with COVID-19 fear (*p* < 0.001). According to our results, university institutions must adopt support mechanisms to alleviate psychological impacts on students during this pandemic, treating it similarly to other disasters. Implications for social work to reduce COVID-19 fear are also discussed.

## 1. Introduction

The World Health Organization [1] declared the coronavirus disease 2019 (COVID-19) to be a pandemic on 11 March 2020. This statement came three months after the appearance of COVID-19 in Wuhan (China) [1]. This situation caused unprecedented political decisions in different countries to manage and confront the health emergency. In Spain, the Spanish Government declared the first State of Emergency on 14 March 2020.

After various extensions and execution of a de-escalation plan, the State of Emergency was finally lifted on 22 June 2020. According to the Instituto Nacional de Estadística (National Statistics Institute) [2], as of May 2020, approximately 45,000 deaths were either suspected of having been caused by COVID-19 or were directly attributed to the disease. In the months following the first wave, different containment measures were developed, applied in the context of a “new normal” [3]. In the main, these measures included social distancing between people, restrictions on social and public gatherings, and capacity restrictions.

Later that year, the incidence rate of the second wave led to the Spanish Government decreeing a second State of Emergency on 25 October 2020. In the early months of 2021, the different Spanish autonomous governments tightened anti-COVID measures to control a third wave of the pandemic. As a result, in February 2021, Spain reported the highest number of accumulated deaths in the same week since the onset of the pandemic. The research carried out collects the data in this phase of the pandemic.

Indeed, since the beginning of the pandemic, there have been over 80,000 deaths from COVID-19 and approximately three and a half million confirmed cases of COVID infection [4]. A vaccination strategy characterized the following spring months. Although vaccination began at the end of December 2020 for the elderly, by mid-June, 84.5% of the population aged over 40 had received at least one dose of a vaccine, while 51% had been fully vaccinated [5]. As of 26 June 2021, the Spanish Government eliminated the obligation to wear a face mask outdoors as long as social distancing of one and a half meters is maintained [6].

Despite these advances, the situation generated by the pandemic has directly affected many aspects of daily life, such as work, education, social relationships, and even future prospects [7]. Studies carried out to date have determined that these changes negatively impact the psychological health of the world population, causing mental problems, such as mood disorders, psychological distress, post-traumatic stress disorder, insomnia, or fear among populations who have undergone quarantine and isolation in different contexts [8,9,10]. Scientific literature has mainly evidenced increased anxiety, depression, and stress disorders [11,12,13,14,15].

The high degree of exposure to the disease COVID-19 and the social and economic effects associated with the pandemic may result in some experiencing a worsening of these psychiatric characteristics (particularly anxiety, depression, and stress), with possible adverse effects on suicidal behaviors [16,17]. The death of close family members and friends, the stigma towards infected people and their families, isolation, physical distancing, changes caused by the digitalization of educational and work activities, unemployment and economic insecurity, information overload, or fear about the lack of social health resources are all critical risk factors that have contributed to the increase in ideation and suicide during the pandemic [18]. Indeed, these factors could peak even after the current pandemic has ended [19].

The context of the pandemic that has been experienced until March 2021 necessitates an exploration of the nature of mental health problems associated with COVID-19 with renewed focus and approaches. The global perspective of social work allows us to interpret mental health as a multidimensional phenomenon in which psychological factors, together with social, cultural, economic, and institutional aspects, play a part. In this sense, the interventions would not only focus on therapeutic or clinical care but on approaches involving community and political action [20]. The interconnection of the biological, psychological, and social dimensions of the COVID-19 pandemic reinforces the need to develop a social work practice directed towards building resilient communities. This aspect implies many factors, from the formulation of institutional policies to individual factors of well-being [21]. Social workers play a critical role in the response to the pandemic for communities that have historically been excluded, but also for those population groups that, for the first time, need support [22].

College students have traditionally been identified as one of the most vulnerable groups in developing mental health problems compared to their counterparts that are not in tertiary education [23,24,25], coinciding with a heightened risk of suicidal behavior [26,27,28]. However, the context of the COVID-19 pandemic has new implications for the psychological effects affecting this collective of young people. In the absence of global studies by country, emerging scientific productions in specific contexts point to a generalized increase in mental health problems in the university community, highlighting anxiety, stress, and depression [27,29,30,31,32,33,34,35,36], as well as suicidal behaviors [37], during the COVID-19 pandemic.

One common element of infectious diseases (such as COVID-19) compared to other conditions is fear [38,39]. Fear is directly associated with its high transmission and mortality, leading people to be unable to think clearly and rationally in the face of COVID-19 [40]. Hoog and colleagues [41] defined fear as an unpleasant emotional state that is triggered by the perception of threatening stimuli. Similarly, Ralph [42] considered fear as an intervening variable between sets of context-dependent stimuli and suites of behavioral responses. A wide range of threats can elicit fear, and given that COVID-19 is affecting our lives in many ways (e.g., on a social, economic, relational, and professional level), fears elicited by COVID-19 may be considerably heterogeneous [43]. Fear can be beneficial or detrimental to mental and physical health during the COVID-19 pandemic. Experiencing fear can increase risk perception, promoting protective behaviors (washing hands and maintaining physical distance, etc.) [44]. For instance, Harper and colleagues [45] found that those individuals that engage in more preventive behaviors do so when they perceive the threat as severe. Taylor and colleagues [46] showed five factors of stress and worries relating to COVID-19: (1) danger and fear of infection, (2) fears about economic consequences, (3) xenophobia, (4) compulsive checking and reassurance seeking, and (5) traumatic stress symptoms. In addition, research carried out in the general population considers stress and anxiety as predictive factors of fear of COVID-19 [13,47,48,49,50].

However, to our knowledge, studies examining1 the risk factors associated with fear of COVID-19 in the college population are rare [51,52,53,54]. To measure fear of COVID-19, Ahorsu and colleagues [47] have developed a brief and valid instrument to capture an individual’s fear of COVID-19 using a five-item Likert-type scale (e.g., “I am afraid of losing my life because of coronavirus-19” and “I cannot sleep because I’m worrying about getting coronavirus-19”). Exceptions to this scarcity are those studies related to issues such as stress, anxiety, anger, or substance use. Given the existing literature, we expect to replicate these same effects in our research. In particular, we expect that anxiety symptoms will be associated with COVID-19 fear. However, our analysis also includes variables that have not been previously analyzed and might act as protective factors against COVID-19 fear (e.g., social support and resilience). We also examine to what extent these variables uniquely, or perhaps in combination with other study variables, might predict COVID-19 fear among a sample of college students. The stress reduction hypothesis [55,56] would suggest that if a pandemic scenario is stressful, then a positive association would be expected between COVID-19 fear, anxiety, and suicide risk, although the causation would remain uncertain. We aimed to determine the level of COVID-19 fear and assess for associations with protective (resilience and social support) and risk (anxiety and suicide risk) factors using data between February and March 2021 from 517 college students in Spain. The present article also presents implications for social work practice that can guide the design of different actions of psychological well-being within university institutions.

## 2. Materials and Methods

### 2.1. Participants

A cross-sectional study was conducted between 1 February and 15 March 2021, in a public university in southeast Spain. The sample included 517 college students enrolled in the university, representing approximately 4.5% of the student population of the college. Participants ranged in age from 18 to 50 years (mean = 21.03; SD = 4.32). There were 409 (79.1%) women and 108 (20.9%) men. This feminized profile is a bias or limitation because respondents come from education and social work degrees, where most students are female. Regarding their relationship status, 262 (50.7%) reported they were not in an intimate relationship, while 255 (49.3%) reported being in a relationship. Most of the students, 78.1%, were in their first year of college, 6% in their second, 0.8% in their third, and 10.8% in their fourth, while the remaining 4.3% stated they were in “other” year of study.

### 2.2. Procedure

Students were eligible to participate if they were enrolled in the university at the time of data collection. No exclusion criteria based on class standing or discipline were applied, with students recruited by an e-mail sent to their institutional address from the university secretary’s office. The Google Forms platform was used for the online survey to record the completed questionnaires. This tool is methodologically suitable for obtaining satisfactory statistical products [57] and has recently been used among college samples to collect sensitive information [40,58]. Before the data collection, the survey was piloted using approximately 30 social work students, assessing their usage experience of different electronic devices (e.g., mobile phones or tablets). The online survey took approximately 10 min to be completed.

Prior to completing the online survey, participants were informed: (1) about the nature and purpose of the study, (2) that participation was anonymous, and the data would be kept confidential, and (3) information provided would be used exclusively for academic purposes by the research staff. If students agreed to participate, they then clicked on a hyperlink to the survey. The research was approved by the university’s Research Ethics Committee (authorization Ref: UALBIO2021/002), and all participants provided online informed consent prior to data collection.

### 2.3. Measures Neither Italics nor Bold Are Necessary

Fear of COVID-19. Ahorsu and colleagues [47] developed the Fear of COVID-19 Scale (FCV-19S) to assess individuals’ anxiety and depressive symptoms that develop due to the COVID-19 outbreak. This instrument consists of a seven-item self-reporting questionnaire with a 5-point Likert scale, ranging from 1 = “strongly disagree” to 5 = “strongly agree”. Total scores ranged from 7 to 35, with higher scores representing a higher level of fear of COVID-19. This instrument has been validated in Spanish among college samples (Martínez-Lorca et al., 2020; α = 0.82). The internal consistency of the scale in the present study was α = 0.84.

Resilience. The Brief Resilient Coping Scale (BRCS) [59] examined tendencies in coping with stress in a highly adaptive manner, using a 5-point Likert scale ranging from 1 = “describes me not at all” to 5 = “describes me very well”. The sum of the items provided a total score ranging between four and twenty. Higher scores indicate higher levels of coping resilience. In this instance, the validated Spanish version of BRCS was used [60]. The scale’s internal consistency in the present study was higher than that reported in the original Spanish validation (α = 0.74. and α = 0.68, respectively).

Social Support. Social support was measured using the Multidimensional Scale of Perceived Social Support (MSPSS) [61]. This instrument is composed of 12 items used to capture, in three sub-scales, perceived social support from friends, relatives, and significant others, scored on a 7-point Likert-type scale ranging from 1 = “completely disagree” to 7 = “completely agree”. The higher the mean score, the more social support an individual perceives. The Spanish version of the MSPSS was used in the current research ([62]: α = 0.82 for friends, α = 0.91 for relatives, and α = 0.94 for significant others). In this study, the internal consistency of the subscale was α = 0.88, α = 0.94, and α = 0.95, for relatives, friends and close relationships, respectively.

Anxiety. The Generalized Anxiety Disorder-7 (GAD-7) [63] was used to assess anxiety symptoms. This instrument comprises seven items, capturing core symptoms of anxiety as outlined in the DSM-IV/DSM-5, using a 4-point Likert-type scale ranging from 0 = “not at all” to 3 = “nearly every day”. The sum of the items provided a total score ranging from 0 to 28, indicating higher scores and higher anxiety levels. In the current research, the internal consistency of the total scale was similar to that reported in the original Spanish validation [64]: α = 0.91 and α = 0.93, respectively.

Risk of Suicide. Suicide risk was examined using the Suicidal Behaviors Questionnaire-Revised (SBQ-R) [65]. The SBQ-R consists of four items that examine the frequency of presentation of suicidal ideation over the past 12 months, communication of suicidal thoughts to others, and attitudes and expectations about current suicide attempts. Items can be analyzed individually and summed up to create a total score ranging between 4 and 21. In this case, the higher the score, the greater the risk of suicidal ideation. According to the authors [49], a score of ≥7 indicates a significant risk of suicidal behavior. Items were adapted from the Spanish version of SBQ-R [66]: α = 0.81. In the current study, Cronbach’s alpha was α = 0.80.

Basic information. The Basic Information section contained the personal information of the respondents, specifically: gender (female and male), age (coded as a continuous variable), whether they were in a serious relationship (yes or no), degree being studied, and year of study. Finally, and based on previous studies measuring risk factors associated with COVID-19 in college students [51], participants were asked about tobacco, alcohol, and cannabis use during the past year; each item was coded as either yes or no.

### 2.4. Data Analysis

In the current research, all statistical analyses were conducted using SPSS (IBM Corp., New York, NY, USA), version 25. Significance was considered at the *p* < 0.05 level. First, each variable or scale was categorized and described (range, *M*, and SD) as the basis for establishing classification criteria. Thus, the dependent variable of the study, *Fear of COVID-19*, has been treated both as a discrete variable in three categories or thirds (low, medium, and high) [51,54,67] as a continuous variable in its original scale format with values ranging between 7 and 35 points. Likewise, the independent variables, *Resilience*, *Social Support*, and *Anxiety*, have been dichotomized through the median point to establish high and low levels in each.

In contrast, the variable “Suicide” has been dichotomized based on the definition of suicide risk proposed by the authors [65]. Second, the chi-square and Student’s *t*-test statistics have been applied to establish associations and statistically significant mean differences between the variables described. However, only the Student’s *t*-test was used for those variables evincing a significant chi-square, after using the Levene test [68] to evaluate the equality of variances. Finally, the correlation coefficients between all the variables have been represented as a scale (Fear of COVID-19, Resilience, Social Support, Anxiety, and Suicide Risk) have been calculated to detect possible multicollinearities and significant relationships between them. Likewise, to achieve our principal objective, a multiple linear regression analysis has been carried out to find the best explanatory model of the dependent variable (COVID-19 fear scale) with the predictive and independent variables (other scales).

## 3. Results

As shown in Table 1, the results describe the five scales used and, additionally, establish the classification criteria and categories of all the variables analyzed. In this way, the descriptive statistics of all the scales vary depending on the range or minimum and maximum values of the results obtained: Fear of COVID-19 (*M* = 18.5; SD = 5.88), Resilience (*M* = 14.4; SD = 2.85), Social Support (*M* = 69.3; SD = 13.53), Anxiety (*M* = 16.8; SD = 5.43), and Suicide (*M* = 6.3; SD = 3.33).

The classification criteria for the dependent variable, or Fear of COVID-19, is based on its three terciles (Tercile 1 = 16; Tercile 2 = 21; and Tercile 3 = 35). Hence, a low level of fear is a score between 7 and 16 (*n* = 188; 36.4%); medium level is between 17 and 21 (*n* = 163; 31.5%); and a high level of fear is between 22 and 35 (*n* = 166; 32.1%). The independent variable of gender was a classification criteria that was easily dichotomized between men (*n* = 108; 20.9%) and women (*n* = 409; 79.1%), the median point for the resilience scales (*M* = 15); Social Support (*M* = 72), Anxiety (*M* = 16), and the definition of risk for the suicide scale, resulted in both low and high values in the scales used to evaluate the independent variables. Thus, the entire sample is divided between those with low (*n* = 318; 61.5%) and high resilience (*n* = 199; 38.5%); low (*n* = 266; 51.5%) and high social support (*n* = 251; 48.5%); low (*n* = 264; 51.1%) and high anxiety (*n* = 253; 48.9%); and at low (*n* = 399; 77.2%) and high risk of suicide (*n* = 118; 22.8%).

Table 2 shows the distribution of the responses concerning the level of fear of COVID-19 with the independent variables of gender and the dichotomized scales of Resilience, Social Support, Anxiety, and Suicide Risk. The only two significant associations according to the chi-squared statistic occur between the three levels of fear of COVID-19 with gender (χ^2^ = 13.0, *p* < 0.01) and Anxiety (χ^2^ = 47.8, *p* < 0.001). In other words, a higher level of fear of the pandemic is linked in a statistically significant way to the female gender and a high degree of anxiety.

In addition, the mean differences in the scale of fear of COVID-19 between men and women and high- and low-anxiety groups have been calculated. First, Levene’s test was applied, which leads us to assume equality of variance in all mean comparisons since the *F* statistic is not significant in any case. Second, the Student’s *t*-test shows significant differences between the means in both genders (*t* = 4.52, *p* < 0.001) and anxiety groups (*t* = 7.51, *p* < 0.001). That is, the mean difference between men and women as well as between low and high levels of anxiety are statistically significant, although it is true that the calculation of the Cohen effect size was medium in the gender variable (*d* = 0.48) and the anxiety about the fear of COVID-19 was relatively large (*d* = 0.66). Nevertheless, these results further reinforce the close link between fear of the pandemic and anxiety specifically.

Before carrying out the multiple linear regression analysis between the dependent variable (COVID-19 fear) and the rest of the four scales and dependent variables, the correlation between all of them has been calculated to detect multicollinearity and which of these could be considered to be significant. Correlation coefficients range from −0.179 (between the Social Support scale and the Suicide scale) and 0.376 (between the scale of Fear of COVID-19 and Anxiety), with which no multicollinearity is detected when there are high correlations (see Table 3). On the other hand, the correlations are only significant between the dependent variable and the Anxiety and Social Support scales. Furthermore, highly significant correlations have been found between Suicide and the Anxiety and Social Support scales, and the latter with Resilience. In any case, the highest and most significant correlation is between Fear of COVID-19 and Anxiety.

Table 4 shows the multiple linear regression analysis results applying an enter method to find the significance of the model and their predictor variables for the dependent variable “COVID-19”. The model is significant (*p*-value < 0.001) and explanatory with a moderately high R (R > 0.444). The two variables that contribute the most to the model for their B coefficients and for their *t* statistics are Anxiety (*t* = 9.791; *p*-value < 0.001) and Suicide Risk (*t* = −4.172; *p*-value < 0.001), although with a different sign, that is, greater anxiety contributes more fear of COVID-19, while the greater risk of suicide implies less fear of the pandemic. Likewise, the predictor variables gender (woman) and Social Support contribute significantly and positively to the explanatory model of fear towards COVID-19. Likewise, the predictor variables gender (female) and social support contribute significantly and positively to the explanatory model of fear of COVID-19. The interaction of all the independent variables within the regression model has led to a greater relevance of low suicide risk and high social support in predicting fear of COVID-19 than was assumed in the correlation matrices.

## 4. Discussion

### 4.1. Comparison with Previous Literature

The COVID-19 epidemic is a public health emergency of global magnitude and, to a greater or lesser extent, poses a challenge to psychological resilience throughout the world in all population groups [69,70]. At first, people, college students not being an exception, encountered an invisible enemy, along with fear, anxiety, and the understanding that, as yet, there was no known cure [67]. The main objective of this research was to evaluate the levels of fear of COVID-19 in a sample of college students and other associated risk and protective factors. The results of the present investigation are consistent with previous studies carried out among the college student population, showing a considerable amount of congruence in responses to COVID-19 [51,54,67]. This further supports the idea that COVID-19 is afflicting college populations at high rates around the world.

Consistent with previous studies [51,52], COVID-19 fear has been associated with a greater risk of anxiety in our sample. Furthermore, according to previous research, the increase in anxiety levels during COVID-19 in college students could be due to the effect of the virus on their education [71], future employment prospects [72], and the gradually increasing distances between people resulting from quarantine measures [29]. Therefore, the present study can add fear of COVID-19 as another risk factor associated with a higher level of anxiety. In addition, current data replicate findings from previous pandemics, specifically the 2009–2010 Swine flu [73] pandemic and the 2015–2016 Zika virus outbreak [74], in which health anxiety was related to increased fear of the current coronavirus pandemic. Despite the fact that COVID-19 fear and anxiety concepts are distinct theoretically and measured in the current research using different scales, their symptoms might overlap, being similar psychological reactions to the COVID-19 epidemic [75]. In order to measure both phenomena in the same instrument, recently Lee and colleagues [76] have created and validated “The Coronavirus Anxiety Scale”, a mental health screener designed to aid researchers in identifying probable cases of dysfunctional anxiety associated with the COVID-19 crisis. Taken together, these findings urge a more in-depth exploration into the association between anxiety and COVID-19 fear for future research among the college population.

In contrast with prior studies [77] showing that higher levels of social support statically decreased COVID-19 fear, social support was positively associated with COVID-19 fear among our sample. This finding might be explained in part because fear can be contagious, thus greater social support can imply more information channels to perceive the fear of COVID-19 [78]. Contrarily, COVID-19 fear had a significant inverse correlation with resilience among college students in the Philippines [79], not being statistically significant among the current sample of students in Spain. In addition, we found that greater social support and resilience reduces anxiety levels during the pandemic. In this way, and given that some of the symptoms of COVID-19 and anxiety are similar, it is recommended to deepen the studies between the independent analyses of these variables, given the benefits of social support and resilience on mental health found by previous authors during the COVID-19 pandemic within different age groups [80].

Notably, COVID-19 fear reduced suicide risk among our sample, explained in part because the possibility of the ambivalent character of “fear” as reverence or respect and terror or horror [81,82]. Both produce anxiety, but respect could be a symptom of a sensible person far removed from risk of suicide. Moreover, the interaction in the regression model analysis of the predictor variable social support, which correlates positively with the fear of COVID-19 (the very likely contagious and social dimension of fear) and negatively with the risk of suicide especially (the protection of social and friendship networks against suicidal ideation), indicates that the risk of suicide might have a negative and significant value in the model. In any case, this hypothesis would require future studies. Finally, regarding demographic variables, compared with men, COVID-19 fear was statically higher among women. This finding is consistent with studies conducted among the general population in Cuba [44], and also consistent with previous scholars that have reported greater psychological vulnerability among women compared to men during the COVID-19 pandemic [15,83]. That women experience more fear than men during COVID-19 may be related to the fact that woman contact people suffering from COVID more often than men do, so for them COVID-19 is a much more real thing. Similarly, Alon and colleagues [84] argued that closures of schools and daycare centers have massively increased childcare needs, which has a particularly large impact on working mothers and sisters among college samples.

### 4.2. Implications for Social Work

COVID-19 has revealed a new scenario for mental healthcare [85,86]. The different academic measures taken in university management worldwide as a direct response to social distancing restrictions and online learning added to the personal, family, and social difficulties resulting from the pandemic. These measures have also contributed to increased levels of anxiety and mental health risk of the college population [30,33,87]. Along these lines, previous studies highlight the need to implement social support and psychological care plans for college students [21]. Scientific evidence globally focuses on students’ mental health (self) care as one of the greatest contemporary and future challenges derived from the COVID-19 pandemic.

In this university context, social workers play an essential role during the current public health emergency. In addition to the teaching role, the professional field is crucial to articulate different resources (social, medical, and psychological) that can cushion the negative impacts of the pandemic [88]. Moreover, authors such as Cifuentes-Faura [89] argue for the need to incorporate the social work perspective into the mental healthcare of college students. Thus, the demand for a greater number of social workers within socio-educational care and guidance counseling teams is justified. Moreover, the support mechanisms for college students during COVID-19 must be implemented under a biopsychosocial model to maintain their resilience and wellbeing, minimizing the effects of the pandemic [67,90].

Specifically, social work must respond to the needs of its students, appealing not only to their commitment to university institutions but also complying with their professional deontology [91]. Different studies carried out with college students studying social work during the COVID-19 pandemic [87,92,93] propose establishing a model that puts the students themselves at the center of the educational organization to strengthen academic certainty and reduce stressors associated with epidemiological exposure to COVID-19.

By spotlighting their concerns and emotional difficulties, student care services can be improved. They can also serve as learning opportunities for future social work professionals [94]. In this sense, authors such as McCarthy and colleagues [95] suggest drawing on the experiences of the students themselves to assimilate the acquisition of the skills needed to handle crisis interventions in the course of their professional life as social workers. Teaching practices would be oriented towards student reflection, collaborative dialogue, and solidarity among the university community members [96].

Moreover, a positive reading of the pandemic can engender opportunities for future graduates in social work to develop greater resilience, a quality that is deemed to be vital in that profession [97]. However, in line with guidelines established by international social work organizations [98], academic institutions face the global challenge of promoting concrete and sustained actions over time that, with the maximum guarantees, address the consequences of COVID-19 in a post-pandemic scenario.

### 4.3. Limitations

There are several limitations to this study. First, mental health symptoms (anxiety and suicide risk) were assessed using self-reported screening tests rather than clinical diagnoses, which may not always be aligned with the objective assessments by mental health professionals. However, these self-reporting instruments have been instrumental in collecting sensitive information during the COVID-19 pandemic in the college population [52,87,99]. In addition, all the instruments used were based on self-assessment questionnaires through online surveys. Second, as the methodology employed was a cross-sectional survey, its design prevents us from determining the causal relationships between the variables analyzed. It should be noted that the current study was carried out exclusively in one university in Spain and that the number of participants is lower than those in studies in other countries [51]. Therefore, although the results are relevant, they are insufficient in terms of potential generalization. Finally, current findings showed how fears and panic depression happened in college students; however, the vaccine has been used and this situation may be changed.

## 5. Conclusions

Among college students in Spain, a high percentage presented fear of COVID-19 (45.1%) with somewhat higher values than a previous study with a similar population [40]. This indicates the need to monitor and track the impact of the fear of the pandemic over time in the population as it can evolve with changing circumstances. It should be noted that the questionnaire was administered in the middle of the third wave of the COVID pandemic in Spain.

Second, the variables most associated with fear of COVID-19 were gender and, especially, anxiety. Female gender and exceptionally high anxiety levels appear to be linked and significantly affect fear of the pandemic, while other variables analyzed such as resilience, social support, and suicide risk did not evince a statistical relationship. The high levels of anxiety established in this work are within the levels found in previous research [100]. The remaining variables merit further analysis with theoretical models, instruments, conceptualizations, analysis strategies, and so forth, different from those used in the present study, to establish other possible relationships with each other.

Given the results obtained, and in line with previous research [29,51,67,87], the mental health of college students is significantly affected when faced with public health emergencies such as COVID-19 [101], and they thus require attention, help, and social support from society, families, and friends. In addition, to meet the challenges of this new situation, it seems necessary for governments and universities to adopt support mechanisms to alleviate its psychological impact on college students.

The implications presented for the practice of social work could be used to formulate socio-educational actions that address the emotional well-being of college students. The evidence from our findings highlights the need to pay special attention to mental health problems. In this sense, the intervention of social work brings together the most individual factors of self-care (stress and anxiety management, promotion of physical well-being), such as the development of social support actions (family network, friends, resource management) that reduce the presence of risk indicators and enhance protective factors.

## Figures and Tables

**Table 1 ijerph-18-08156-t001:** Description and classifications of all variables (*N* = 517).

	Values or Min–Max	Mean	SD	Criterial	Classifications (*n*)
Gender	1 = Male 2 = Female	-	-	Gender	Male (108) Female (409)
Level	1 = First year 2 = Remainder			Level	First (404) Remainder (113)
Fear of COVID-19	7 = min. 35 = max.	18.5	5.88	Tercile 1 = 16 Tercile 2 = 21 Tercile 3 = 35	Low—tercile 1 (188) Medium—tercile 2 (163) High—tercile 3 (166)
Resilience	4 = min. 20 = max.	14.4	2.85	Median = 15	Low or ≤ Median (318) High or > Median (199)
Social Support	13 = min. 84 = max.	69.3	13.53	Median = 72	Low or ≤ Median (266) High or > Median (251)
Anxiety	7 = min. 28 = max.	16.8	5.43	Median = 16	Low or ≤ Median (264) High or > Me (253)
Suicide	4 = min. 21 = max.	6.3	3.33	[65] ≥ 7	No risk or < 7 (399) Suicide Risk or ≥ 7 (118)

**Table 2 ijerph-18-08156-t002:** Distribution of COVID-19 fear level responses of students.

	Level of COVID-19 Fear		
	Low (*n* = 188)	Medium (*n* = 163)	High (*n* = 166)
Gender (female), % (*n*)	71.3 (134) **	80.4 (131) **	86.7 (144) **
Level (First year), % (*n*)	81.4 (153)	79.1 (129)	73.5 (122)
Resilience (high), % (*n*)	36.7 (69)	36.2 (59)	42.8 (97)
Social Support (high), % (*n*)	44.7 (84)	47.9 (78)	53.6 (89)
Anxiety (high), % (*n*)	29.8 (56) ***	54.0 (88) ***	65.7 (106) ***
Suicide Risk, % (*n*)	23.9 (45)	21.5 (35)	22.9 (38)

** *p* < 0.01, *** *p* < 0.001 (χ^2^ test).

**Table 3 ijerph-18-08156-t003:** Correlations among scales.

	COVID-19 Fear	Anxiety	Social Support	Suicide Risk	Resilience
COVID-19 Fear					
Anxiety	0.376 **				
Social Support	0.103 *	−0.095 *			
Suicide Risk	−0.044	0.346 **	−0.179 **		
Resilience	−0.012	−0.063	0.368 **	−0.037	

* *p* < 0.05, ** *p* < 0.01.

**Table 4 ijerph-18-08156-t004:** Model of multiple linear regression analysis (dependent variable = COVID-19 fear).

	B	S.E.	T	Sig. T
Constant	9.010	1.746	5.161	0.000
Gender	1.635	0.593	2.756	0.006
Anxiety	0.458	0.047	9.791	0.000
Social support	0.047	0.019	2.445	0.015
Suicide Risk	−0.389	0.093	−4.172	0.000
Resilience	−0.055	0.089	−0.625	0.532
R = 0.446; R^2^ = 0.199 (F = 25.28; *p* < 0.001).				

## Data Availability

Muyor Rodríguez, Jesús; Caravaca Sánchez, Francisco; Fernández-Prados, Juan Sebastián (2021), “COVID-19 Fear, Resilience, Social Support, Anxiety, and Suicide among College Students in Spain”, Mendeley Data, V1, doi:10.17632/phpbf46xr8.1.

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
