# Peer review of "COVID-19 Fear, Resilience, Social Support, Anxiety, and Suicide among College Students in Spain"

_ijerph, 2021, doi:10.3390/ijerph18158156_

Round 1

Reviewer 1 Report

The article submitted to the IJERPH journal (ISSN 1660-4601), presents a high quality and rigorous methodological work. Its results are well presented and its conclusions are well established.
However, there are elements that could be improved:
-It would be advisable to expand the contributions from Social Work and its connection with the article and its main findings.... It would be advisable to refer to the most recent bibliography of social work and to expand in the Introduction section.
-In relation to this, it would be necessary to connect the new Social Work contributions in the Introduction with the conclusions section, so that everything is well structured and connected. The Conclusions section should be expanded a little more. 

Author Response

We thank the editors and reviewers of the International Journal of Environmental Research and Public Health for the thorough reviews and constructive feedback that guided the revision of our manuscript.  We hope that you agree that it has been improved by our revisions.

Below, please find point by point responses to the comments from the four reviewers.

REVIEWER 1

Reviewer (1.1):. It would be advisable to expand the contributions from Social Work and its connection with the article and its main findings.... It would be advisable to refer to the most recent bibliography of social work and to expand in the Introduction section

Authors (1.1): The comments of the review are taken into account and an expansion of the specific bibliographic review of Social Work is carried out. With this, the introduction section is expanded by incorporating the contributions of Social Work to the field of study proposed in the article (please see lines 70 to 84)

Reviewer (1.2): In relation to this, it would be necessary to connect the new Social Work contributions in the Introduction with the conclusions section, so that everything is well structured and connected. The Conclusions section should be expanded a little more.

Authors (1.2): Following the reviewer's comment, the final conclusions are expanded with the implications for the practice of Social Work (lines 409 to 415). These implications are finally connected through the introduction, discussion and conclusions sections.

Reviewer 2 Report

I have attached my comments in the attached paper. This paper is well-written and timely, and is commendably focused and brief. I believe it should be published. The authors are researching an important topic and, while aware of the study;'s limitations, they are offering research data which supports the emerging consensus about  COVID and its impact on university students. This will contribute to the growing literature in this area. I have suggested that the authors might consider repeating the study with only the data fro First Year students, which constitutes 78% of their sample. This could provide insightful data on the impact of COVID on those who began university studies during the pandemic. 

Author Response

We thank the editors and reviewers of the International Journal of Environmental Research and Public Health for the thorough reviews and constructive feedback that guided the revision of our manuscript.  We hope that you agree that it has been improved by our revisions.

Below, please find point by point responses to the comments from the four reviewers.

REVIEWER 2

Introduction

Reviewer (2.1): The text states that approximately 30,000 people died, and then adds, “only approximately 45,000 of those deaths . . .” There must be an error here. Should the numbers be reversed?

Authors (2.1): Following your recommendation the paragraph was revised and modified and now reads: “According to the Instituto Nacional de Estadística (National Statistics Institute) [2], as of May 2020, approximately 45,000 deaths were either suspected of having been caused by COVID-19 or were directly attributed to the disease.

Reviewer (2.2): I wonder if “probable” adverse effects on suicidal behaviors could more

accurately described as “possible,” or “likely” adverse effects, to avoid this

sounding like a definite prediction? Something for the authors to consider.

Authors (2.2): We thank R2 for this comment, the sentence was re-writing and now reads: “The high degree of exposure to the disease COVID-19 and the social and economic effects associated with the pandemic may result in some experiencing a worsening of these psychiatric characteristics (particularly anxiety, depression, and stress) with possible adverse effects on suicidal behaviors [16,17]”

Reviewer (2.3): “anger: rather than “angry,” which is an adjective.

Authors (2.3): We thank R2 for this comment, the sentence was re-writing and now reads: “Exceptions to this scarcity are those studies related to issues such as stress, anxiety, anger, or substance use.

Reviewer (2.4): 78.1% of the sample were I their first year of study. This raises questions

about the extent to which the anxiety was related to being in First Year or the extent to which first years students are more susceptible to the mental health issues being researched due to age, lack of university experience, adjusting to being a University student while remaining at home, or accessing learning through distance learning etc. I wonder, given the high percentage of First Year students in the study, if there would be merit in running the study with only the First year students, or to compare, using Ttests the First year students and the other 21% of students in the study. My concern here would be lack of power to run the T-tests, but a statistician will be able to make that decision. There could also be merit in doing research with students in their final year only ( or conducting the research with the results of this group on their own) to see if the prospect of leaving university and establishing a career in the context of the pandemic is related to an increase in anxiety stress and fear, and how this might be alleviated by age, coping mechanisms (resilience), relationship status or family support

Authors (2.4): Dear reviewer, the dichotomized year variable (first year and the rest) has been entered in Table 1 and Table 2. There is no association with our study variable or dependent variable (fear of COVID-19) according to the chi-square statistic shown in Table 2. We understand the interest and concern of the reviewer, but establishing an analysis of how the level or course influences the predictive variables and scales would mean the development of another line of research that we will take into account for other studies.

Table 1

Descriptive and classifications of all variables (N = 517)

Values or Min-Max

Mean

SD

Criterial

Classifications (n)

Gender

1 = male

2 = female

-

-

Gender

Male (108)

Female (409)

Level

1 = First year

2 = Rest

Level

First (404)

Rest (113)

Fear of COVID-19

7 = min.

35 = max.

18.5

5.88

Tercile 1 = 16

Tercile 2 = 21

Tercile 3 = 35

Low- tercile 1 (188)

Medium - tercile 2 (163)

High - tercile 3 (166)

Resilience

4 = min.

20 = max.

14.4

2.85

Median = 15

Low or ≤ Median (318)

High or > Median (199)

Social support

13 = min.

84 = max.

69.3

13.53

Median = 72

Low or ≤ Median (266)

High or > Median (251)

Anxiety

7 = min.

28 = max.

16.8

5.43

Median = 16

Low or ≤ Median (264)

High or > Me (253)

Suicide

4 = min.

21 = max.

6.3

3.33

(Osman et al., 2001) ≥ 7

No risk or < 7 (399)

Suicide risk or ≥ 7 (118)

Table 2

Distribution of COVID-19 Fear level responses to students’ measures.

Level of COVID-19 Fear

Low

(n = 188)

Medium

(n = 163)

High

(n = 166)

Gender (female), % (n)

71.3 (134)**

80.4 (131)**

86.7 (144)**

Level (First year), % (n)

81.4 (153)

79.1 (129)

73.5 (122)

Resilience (high), % (n)

36.7 (69)

36.2 (59)

42.8 (97)

Social support (high), % (n)

44.7 (84)

47.9 (78)

53.6 (89)

Anxiety (high), % (n)

29.8 (56)***

54.0 (88)***

65.7 (106)***

Suicide risk, % (n)

23.9 (45)

21.5 (35)

22.9 (38)

**p < .01 ***p < .001 (χ2 test)

Reviewer (2.5): The full stop should come after (Killgore, et al . . .) and not before

Authors (2.5): Thanks you for the catch, the reference has been updated following the references style for the Journal. Apologies for the misunderstand. The sentence now reads: “The COVID-19 epidemic is a public health emergency of global magnitude and, to a greater or lesser extent, poses a challenge to psychological resilience throughout the world in all population groups [65,66].”.

Reviewer (2.6): It is true that the lack of clinical diagnosis may be a limitation, but self-report in this context must also carry a lot of weight, as it reveals the students’ self-perception of the effects of COVID on their well-being

Authors (2.6): Following your recommendation the revised draft includes information about previous research that used self-reported information for measuring COVID-19 variables satisfactorily during the pandemic. Thus, the first limitation now reads:

There are several limitations to this study. First, mental health symptoms (anxiety and suicide risk) were assessed using self-reported screening tests rather than clinical di-agnoses, which may not always be aligned with the objective assessments by mental health professionals. However, these self-reporting instruments have been instrumental in collecting sensitive information during the COVID-19 pandemic in the college population [48,75,87]”

Reviewer 3 Report

This a cross-sectional study of 517 college students (79.1% women and 20.9% men) from a public university in the southeast of Spain. The authors used the fear of COVID-19 scale (FCV-19S) questionnaire and questions related to resilience, social support, anxiety, and suicide risk levels. The major result was the factor of anxiety was positive association and effect with COVID-19 fear (p < .001). They suggested that “University institutions must adopt support mechanisms to alleviate the psychological impact on students during this pandemic, treating it similarly to other disasters”

I have some comments.

  1. The study was done during February 1 to March 15, 2020. The Covid-19 was just pandemic. To date the situation was changed very much. The introduction, and discussion have to rewrite focused on at beginning when a new disease happened, also I wished an additional information about the changes in 2021.
  2. The study participants were not randomly selected, and major in women and the first-year college students. Thus, the limitation and inference bias must discuss.
  3. Table 3 please show the p-values of t-test, not F and sig.
  4. I have tried to download supplementary file(s), but fail. I wish to read the questionnaires.

Author Response

We thank the editors and reviewers of the International Journal of Environmental Research and Public Health for the thorough reviews and constructive feedback that guided the revision of our manuscript.  We hope that you agree that it has been improved by our revisions.

Below, please find point by point responses to the comments from the four reviewers.

REVIEWER 3

Reviewer (3.1): The study was done during February 1 to March 15, 2020. The Covid-19 was just pandemic. To date the situation was changed very much. The introduction, and discussion have to rewrite focused on at beginning when a new disease happened, also I wished an additional information about the changes in 2021

Authors (3.1): The introduction has been modified by adding information on the evolution of the pandemic and the main changes produced in the Spanish context. Following his recommendation, the following text has been introduced:

A vaccination strategy characterized the following spring months. Although vaccination began at the end of December 2020 for the elderly, by mid-June, 84.5% of the population aged over 40 has received at least one dose of a vaccine, while 51% have been fully vac-cinated [5]. As of June 26, 2021, the Spanish Government of Spain eliminated the obliga-tion to wear a face mask outdoors as long as social distancing of one and a half meters is maintained [6].

Reviewer (3.2): The study participants were not randomly selected, and major in women and the first-year college students. Thus, the limitation and inference bias must discuss

Authors (3.2): As suggested, further information has been added regarding our feminized profile sample in the “Participants” subsection. Now reads:

“Participants ranged in age from 18 to 50 years (Mean=21.03; SD=4.32). There were 409 (79.1%) women and 108 (20.9%) men. This feminized profile is a bias or limitation be-cause respondents come from education and social work degrees, where most students are female”

Reviewer (3.3): Table 3 please show the p-values of t-test, not F and sig.

Authors (3.3): As requested, table 3 has been modified.

n

Mean (7-35)

SD

t

Sig

Gender

  Male

108

16.3

5.70

4.52*

.000

  Female

409

19.1

5.79

Anxiety

  Low

264

16.7

5.70

7.51*

.000

  High

253

20.4

5.45

*p < .001

Reviewer (3.4): I have tried to download supplementary file(s), but fail. I wish to read the questionnaires.

Authors (3.4): Dear reviewer, you can find the whole questionnaire in the next link: https://forms.gle/wQP5gojSeChrjhKu5

Reviewer 4 Report

 I totally agree with the authors that we need more research on the psychological aspects of the recent pandemic, so such studies as presented in this manuscript should be appreciated highly. The paper has some potentials, but here are some issues that needed more attention, and hopefully, the manuscript should be re-write and data re-analyzed. The most important thing which casts a grim shadow on the whole study is that the aims /objectives of the study are unclear. First of all, the authors decline to state them explicitly. And it is difficult to infer from the reading of an introduction and the rest of the manuscript what is all about. 

    1. Is it looking for risk and protective factors as predictors of fear of COVID-19 [77-78, 85-87]? In that case, fear of COVID-19 is a dependent (explained) variable [166-167], and the proper statistical analysis is regression analysis. But this analysis has not been performed.
    2. Or maybe the question is what are the possible associations between some psychological variables and fear of COVID-19, regardless of the direction of causality [82-85]? In that case, the status of fear of COVID-19 as a variable is the same as other variables. Calculation of correlation coefficients, also of chi-square indicators are acceptable procedures. But there was no need to convert continuous variables into categorical ones.
    3. However, in further analysis, fear of COVID-19 is presented as an independent variable, when the authors divide it into three terciles [The classification criteria for the independent variable or Fear of COVID-19 is based 185 on its three terciles (Tercile 1 = 16; Tercile 2 = 21, and Tercile 3 = 35). Hence, a low level of 186 fear is a score between… lines 185-187].  
    4. The only significant results reported by the authors are those related to gender (this might not be an entirely valid result due to the unbalanced proportion of genders in the sample) and to the anxiety (which is so obvious – fear and anxiety usually are correlated with each other!).  

 I am going to give also comments on some details of the manuscript.

  1. Introduction
    1. The authors write that College students have traditionally been identified as one of the most vulnerable 66 groups in developing mental health problems. But they didn’t explain why is so. Are they more vulnerable than other age groups? Or students at other education levels, e.g. high school level? Or other professional groups? What are the mechanisms underlying that specific vulnerability?
    2. The authors didn’t provide any theoretical background justifying the selection of variables and what relationships among them might be. There are many psychological theories concerning such problems as stress, emotions, well-being etc, which might be a good source of hypotheses (which are not formulated in this study).
    3. There are no theoretical definitions of examined variables. Specifically, how do the authors define “anxiety factors” and “COVID-19 fear”? Distinctions between them are not explained.

2. Materials and Methods

Participants. Two characteristics of the sample are of concern. The wide disparity of age (18 -50 years) and an unbalanced gender proportion (80% to 20%). How can these factors influence the results? Besides, age mean and standard deviation values should be provided. 

Measures. The authors should present the instruments (all items) as supplement materials. Also, one or two exemplary items should be included in the description of the measures; it would help in a better understanding of the variables.

3. Data Analysis

    1. Could the authors explain why they convert the continuous variables provided by the instruments into categorical ones? Such a procedure results in losing much information and poses restrictions on the availability of statistical procedures. If there were reasons for that, they must be stated explicitly.   

4. Results

    1. The authors refer to Table 1, which is not included in the manuscript.
    2. Descriptive statistics of all variables should be provided in the table, i.e., means, standard deviations, and intercorrelations. It would inform, for example, how much anxiety and fear variables are correlated, as they are conceptually similar (see comments above).  
    3. For this kind of data, a regression analysis is more suitable, with continuous variables employed. Definitely, the authors should re-analyze the data.
  1. 5. Discussion
    1. Actually, there is no discussion of the results. Two paragraphs of Comparison with previous literature could not serve as a proper discussion.
    2. The authors write that The main objective of this research was to evaluate the levels of fear of COVID-19 in a sample of college students, but this aim was impossible to obtain with their approach.
  2. Supplement materials
    1. It is the main manuscript copied.

Author Response

We thank the editors and reviewers of the International Journal of Environmental Research and Public Health for the thorough reviews and constructive feedback that guided the revision of our manuscript.  We hope that you agree that it has been improved by our revisions.

Below, please find point by point responses to the comments from the four reviewers.

REVIEWER 4

Reviewer (4.1):. The most important thing which casts a grim shadow on the whole study is that the aims /objectives of the study are unclear.

Authors (4.1): Following your recommendation, the aim of the current research has been introduced more clearly at the end of the introduction, specifically:

We aimed to determine the level of COVID-19 fear and assess for associations with protective (resilience and social support) and risk (anxiety and suicide risk) factors using data from 517 college students in Spain. The present article also presents implications for social work practice that can guide the design of different actions of psychological well-being within university institutions

Reviewer (4.2): First of all, the authors decline to state them explicitly. And it is difficult to infer from the reading of an introduction and the rest of the manuscript what is all about. Is it looking for risk and protective factors as predictors of fear of COVID-19 [77-78, 85-87]? In that case, fear of COVID-19 is a dependent (explained) variable [166-167], and the proper statistical analysis is regression analysis. But this analysis has not been performed.

Authors (4.2): Dear reviewer, following your advices Two tables have been added, one with the correlations (table 4) and the other with a multiple linear regression analysis (table 5) with their respective comments.

  • “Before carrying out the multiple linear regression analysis between the dependent variable (fear of COVID-19) and the rest of the four scales and dependent variables, the correlation between all of them has been calculated to detect if there is multicollinearity and which of these could be considered to be significant. On the one hand, the correlation coefficients range from -0.179 (between the support scale and the suicide scale) and 0.376 (between the scale of fear of COVID 19 and anxiety), with which no multicollinearity is detected when there are no very high correlations (see Table 4). On the other hand, the correlations are only significant between the dependent variable and the anxiety and support scales. Furthermore, highly significant correlations have been found between sui-cide and the anxiety and support scales, and the latter with resilience. In any case, the highest and most significant correlation is between fear of COVID-19 and anxiety”.

Table 4

Correlations among scales.

COVID-19 fear

Anxiety

Social support

Suicide risk

Resilience

COVID-19 fear

Anxiety

.376**

Social support

.103*

-.095*

Suicide risk

-.044

.346**

-.179**

Resilience

-.012

-,063

.368**

-.037

*p < .05 **p < .01

  • “Table 5 shows the multiple linear regression analysis results applying a backward stepwise method to find the best explanatory model for the dependent variable "fear of COVID-19". Two explanatory models have been obtained. One uses the four scales as predictors, while in the other, the resilience scale disappears because it is not significant. Both models are significant (p-value <0.001) and explanatory with a moderately high R (R> 0.43). In the two explanatory models, the two variables that contribute the most to the two models both for their B coefficients and for their t statistics are anxiety (t = 10.568; p-value <0.001) and suicide risk (t = -4.171; p-value <0.001 ), although with a different sign, that is, greater anxiety contributes more fear of COVID-19, while the greater risk of suicide implies less fear of the pandemic”.

Table 5. Models of Multiple Linear Regression Analysis.

Model I

Model II

B

S.E.

T

Sig. T

B

S.E.

T

Sig. T

Constant

9.600

1.738

5.524

0.000

8.895

1.538

5.782

0.000

Anxiety

0.484

0.046

10.517

0.000

0.486

0.046

10.568

0.000

Social support

0.055

0.019

2.926

0.004

0.049

0.018

2.801

0.005

Suicide risk

-0.314

0.076

-4.127

0.000

-0.317

0.076

-4.171

0.000

Resilience

-0.077

0.089

-0.873

0.383

R; R2 (F; Sig.)

R=.435; R2= .189 (F=29.87; p< .001)

R=.434; R2=.188 (F=39.59; p< .001)

Reviewer (4.3): Or maybe the question is what are the possible associations between some psychological variables and fear of COVID-19, regardless of the direction of causality [82-85]? In that case, the status of fear of COVID-19 as a variable is the same as other variables. Calculation of correlation coefficients, also of chi-square indicators are acceptable procedures. But there was no need to convert continuous variables into categorical ones.

Authors (4.4): Please see the above comment

Reviewer (4.4): However, in further analysis, fear of COVID-19 is presented as an independent variable, when the authors divide it into three terciles [The classification criteria for the independent variable or Fear of COVID-19 is based 185 on its three terciles (Tercile 1 = 16; Tercile 2 = 21, and Tercile 3 = 35). Hence, a low level of 186 fear is a score between… lines 185-187].

Authors (4.4): Apologies for the misunderstanding, the sentence was modified and now reads: “The classification criteria for the dependent variable or Fear of COVID-19 is based on its three terciles (Tercile 1 = 16; Tercile 2 = 21, and Tercile 3 = 35)”

Reviewer (4.5): The only significant results reported by the authors are those related to gender (this might not be an entirely valid result due to the unbalanced proportion of genders in the sample) and to the anxiety (which is so obvious – fear and anxiety usually are correlated with each other!).

Authors (4.5):

  • The reality of the profile of social work and social education students (most of those who have responded) is highly feminized, as highlighted in the description of the sample, which generates this imbalance, although it is more in line with reality. In any case, the subsample of men is greater than one hundred cases, which allows these statistical calculations to be made [we have included a clarification of the provenance or studies of the sample in the “participants” section
  • A difference is made between fear and anxiety, following the explanations by Corchs et al. (2015) who point out that for the first time a difference is made between fear and anxiety in the DSM V [This reference is included and a phrase is introduced referring to DSM V.
    • Reference: Hince, D. A., Davies, S. J., Bernik, M., & Hood, S. D. (2015). Evidence for serotonin function as a neurochemical difference between fear and anxiety disorders in humans?. Journal of Psychopharmacology, 29(10), 1061–1069. https://doi.org/10.1177/0269881115590603

Reviewer (4.6): The authors write that College students have traditionally been identified as one of the most vulnerable 66 groups in developing mental health problems. But they didn’t explain why is so. Are they more vulnerable than other age groups? Or students at other education levels, e.g., high school level? Or other professional groups? What are the mechanisms underlying that specific vulnerability?

Authors (4.6): As suggested, more detailed information has been added regarding higher vulnerability among college students samples compared with general population, specifically: “College students have traditionally been identified as one of the most vulnerable groups in developing mental health problems compared to their counterparts not in ter-tiary education [23–25], coinciding with a heightened risk of suicidal behavior [26–28]”.

Reviewer (4.7): The authors didn’t provide any theoretical background justifying the selection of variables and what relationships among them might be. There are many psychological theories concerning such problems as stress, emotions, well-being etc, which might be a good source of hypotheses (which are not formulated in this study).

Authors (4.7): As suggested, further information has been added regarding psychological theories concerning such problems as stress, specifically: “The stress reduction hypothesis [51,52] would suggest that if a pandemic scenario is stressful, then a positive association would be expected between COVID-19 fear, anxiety, and suicide risk, although the causation would remain uncertain.”.

Reviewer (4.8): There are no theoretical definitions of examined variables. Specifically, how do the authors define “anxiety factors” and “COVID-19 fear”? Distinctions between them are not explained.

Authors (4.8): In this revised version of the manuscript two modifications has been conducted:

  • Page 2: In line with the Generalized Anxiety Disorder-7, in the current research we are measuring “anxiety symptoms” instead of “anxiety factors”. Thus, the sentence in the introduction section has been modified and now reads: “Given the existing literature, we expect to replicate these same effects in our research. In particular, we expect that anxiety symptoms will be associated with COVID-19 fear”
  • Page 2 and page 3: Despite the fact that to our knowledge there is no official theoretical definition of COVID-19 fear, we have added in the introduction section in and Measures section information on the stressors and worries associated with COVID-19 fear proposed by different authors to better contextualize the analysed variable. Specifically:
    • Introduction (page 2):A wide range of threats can elicit fear, and given that COVID-19 is affecting our lives in many ways (e.g., on a social, economic, relational, and professional level), fears elicited by COVID-19 may be considerably heterogeneous [41]. Taylor and colleagues [42] showed five factors of stress and worries relating to COVID-19: (1) danger and fear of infection, (2) fears about economic consequences, (3) xenophobia, (4) compulsive checking and reas-surance-seeking, and (5) traumatic stress symptoms.”
    • “Fear of COVID-19 subsection: “Ahorsu and colleagues [43] developed the Fear of COVID-19 Scale (FCV-19S) to assess in-dividuals' anxiety and depressive symptoms that develop due to the COVID-19 outbreak”

Materials and Methods

Reviewer (4.9): Participants. Two characteristics of the sample are of concern. The wide disparity of age (18 -50 years) and an unbalanced gender proportion (80% to 20%). How can these factors influence the results? Besides, age mean and standard deviation values should be provided.

Authors (4.9):

  • The age disparity has raised the mean to 21.03 years with a SD of 4,324 when the usual age of first-year students corresponds between 18-19 years [we have incorporated these descriptive data (mean and SD) in the subsection of “participants")
  • The reality of the profile of social work and social education students (most of those who have responded) is highly feminized, as highlighted in the description of the sample, which generates this imbalance, although it is more in line with reality. In any case, the subsample of men is greater than one hundred cases, which allows these statistical calculations to be made [we have included a clarification of the provenance or studies of the sample in the “participants” section

Reviewer (4.10): Measures. The authors should present the instruments (all items) as supplement materials. Also, one or two exemplary items should be included in the description of the measures; it would help in a better understanding of the variables.

Authors (4.10): Dear reviewer, you can find the whole questionnaire in the next link: https://forms.gle/wQP5gojSeChrjhKu5

Reviewer (4.11): Could the authors explain why they convert the continuous variables provided by the instruments into categorical ones? Such a procedure results in losing much information and poses restrictions on the availability of statistical procedures. If there were reasons for that, they must be stated explicitly.

Authors (4.11):

As indicated in the subsection “Data analysis” The data analysis strategy is motivated mainly by the review of previous studies. The definition of the intervals of the three tertiles is described in the text, following the same methodological strategy as previous authors:

  • Gritsenko, V., Skugarevsky, O., Konstantinov, V., Khamenka, N., Marinova, T., Reznik, A., & Isralowitz, R. (2020). COVID 19 Fear, Stress, Anxiety, and Substance Use Among Russian and Belarusian University Students. International Journal of Mental Health and Addiction. https://doi.org/10.1007/s11469-020-00330-z.

The dichotomized division of the high and low suicide risk scales follows the definition established by the authors (Osman et al., 2001) as with other psychological variables (Gritsenko et al., 2020).

Reviewer (4.12): The authors refer to Table 1, which is not included in the manuscript.

Authors (4.12): We apologize for the error. Table 1 has been added to the main document.

Reviewer (4.13): Descriptive statistics of all variables should be provided in the table, i.e., means, standard deviations, and intercorrelations. It would inform, for example, how much anxiety and fear variables are correlated, as they are conceptually similar (see comments above).

Authors (4.13): Dear reviewer, requested information has been added, please see tables 1 and 4

Reviewer (4.14): For this kind of data, a regression analysis is more suitable, with continuous variables employed. Definitely, the authors should re-analyze the data.

Authors (4.14): Dear reviewer, requested information has been added, please see table 5

Reviewer (4.15): Actually, there is no discussion of the results. Two paragraphs of Comparison with previous literature could not serve as a proper discussion.

Authors (4.15): As suggested, further information has been added in the discussion section in the section named as 4.1 “Comparison with previous literature”: “In addition, current data replicate findings from previous pandemics, specifically, the 2009–2010 Swine flu [69] pandemic and the 2015–2016 Zika virus outbreak [70], in which health anxiety was related to increased fear of the current coronavirus pandemic. In con-trast with prior studies [71] showing that higher levels of social support statically decrease COVID-19 fear, social support was not associated with COVID-19 fear among our sample. Similarly, COVID-19 fear had a significant inverse correlation with resilience among col-lege students in the Philippines [72], not being statistically significant among the current sample of students in Spain”.

Reviewer (4.16): The authors write that The main objective of this research was to evaluate the levels of fear of COVID-19 in a sample of college students, but this aim was impossible to obtain with their approach.

Authors (4.16): Dear reviewer, as you can see in the main document, the main aim of the research has been re-formulated and two new tables (tables 4 and 5) has been added.

We understand that with the analysis suggested by the reviewer (correlations and multiple linear regression) it is possible to achieve the proposed objectives more clearly.

  • “We aimed to determine the level of COVID-19 fear and assess for associations with protec-tive (resilience and social support) and risk (anxiety and suicide risk) factors using data from 517 college students in Spain.
  • Please, see tables 4 and 5

Round 2

Reviewer 3 Report

As I mentioned “The study was done during February 1 to March 15, 2020. The Covid-19 was just pandemic. To date the situation was changed very much….”, the authors responded that “A vaccination strategy characterized the following spring months….  “. However, not only the vaccine issue, authors must state their study began at the early phase of this pandemic. The goal of this research revealed, at beginning of a pandemic, how fears and panic depression happened in college students. I suggest to emphasize this point in the section of line 37, lines 70-81, lines 109-117, and adds this point in the limitation (the findings showed the younger people faced a pandemic at beginning phase; however, the vaccine used and this situation may be changed.).

Author Response

We thank the editors and reviewers of the International Journal of Environmental Research and Public Health for the thorough reviews and constructive feedback that guided the revision of our manuscript.  We hope that you agree that it has been improved by our revisions.

Below, please find point by point responses to the comments from the two reviewers.

REVIEWER 3

Reviewer (3.1): As I mentioned “The study was done during February 1 to March 15, 2020. The Covid-19 was just pandemic. To date the situation was changed very much….”, the authors responded that “A vaccination strategy characterized the following spring

months…. “. However, not only the vaccine issue, authors must state their study began at the early phase of this pandemic. The goal of this research revealed, at beginning of a pandemic, how fears and panic depression happened in college students. I suggest to emphasize this point in the section of line 37, lines 70-81, lines 109-117

Authors (3.1): An error was detected and corrected in the revised version of the manuscript. The data collection year was in 2021. In addition, the reviewer's comments are included in the aforementioned lines to emphasize the specific data collection period of the study.

Reviewer (3.2): and adds this point in the limitation (the findings showed the younger people faced a pandemic at beginning phase; however, the vaccine used and this situation may be changed.)

Authors (3.2): The reviewer's appreciation is included as a study limitation. Specifically: “Finally, current findings showed how fears and panic depression happened in college students; however, the vaccine used and this situation may be changed”.

Reviewer 4 Report

Dear Authors, thank you for your answers to my questions. I appreciate your effort to make your paper more scholarly sound. Almost all issues raised in my previous review have been addressed reasonably, but some problems remain unresolved.

Sorry for not being clear enough in my previous review about your statistical analysis. I intended to draw your attention to the fact that you had nice data obtained with your measurement scales which allowed you to perform more advanced analysis (e.g., regression analysis). The χ2 test would show you simple associations between two nominal variables only (e.g., gender and fear, as in Table 2), while in regression analysis, you can put all your variables (including gender) together in the model, which is a more advisable approach. In my opinion, there is no reason to convert continuous (numeric) variables into nominal (dichotomized) ones; at least, I can’t see any reason provided explicitly in your manuscript. But you did so, and there is no harm in this, unless we look at the tables more carefully. Table 2 shows a higher level of fear of the pandemic is linked in a statistically significant way to the female gender and a high degree of anxiety” [252, 253,254]. 

  1. The next table (Table 3) shows the results of the t-test: the mean difference between men and women as well as between low and high levels of anxiety are statistically significant [264,265], which are actually the same results as those in Table 2 (concerning two variables – gender and anxiety), but with a piece of additional information about effect size. It is easy to see some redundancy in reporting your results, which is also present in Table 5. Table 5 shows the results of the regression analysis that you performed to test two models. Each of them produces the same results; obviously, the Model II – as such as it is - is redundant. It would be more beneficial if you put in the other model an additional variable, such as gender, and apply Enter method to get more valuable information about possible predictors.   

  1. But there are more problems with your findings. What you got with regression analysis (Table 5) is incompatible with your earlier analyses (Tables 2 & 4). I mean the finding of the role of suicide risk in the prediction of fear of COVID: In the two explanatory models, the two variables that contribute the most to the two models both for their B coefficients and for their t statistics are anxiety (t = 10.568; p-value <0.001) and suicide risk (t = -4.171; p-value <0.001), although with a different sign, that is, greater anxiety contributes more fear of COVID-19, while the greater risk of suicide implies less fear of the pandemic. How do you explain such a high association between suicide risk and fear of COVID in regression analysis, while you demonstrated no associations between these variables in former analyses (Table 2 - χ2 test & Table 4 – correlations)?

  1. In the section named “Discussion,” there is no attempt to discuss your results. You should explain your findings concerning:
    1. The positive association between fear of COVID-19 and anxiety.

It would help enormously if you give clear explanations in the Introduction how you understand the concepts: fear and anxiety, and give examples of items of the employed scales (anyway, thank you for the link to your questionnaires, in Spanish though). Although these concepts are distinct theoretically, their symptoms overlap. So, perhaps what you measure with an anxiety scale there are just symptoms of the fear of the COVID? Is it possible that these instruments measure the same phenomenon? Whatever the truth is, you should discuss possible explanations; reference to studies by other scholars is not enough.

    1. Negative and significant correlation between suicide risk and fear of COVID (if there is a correlation!)
    2. Lack of expected associations between social support & resilience and fear of COVID
    3. Higher level of fear of COVID in women

My guess is that women contact people suffering from COVID more often than men do, so perhaps for them, COVID is a much more real thing. Is it possible? Or can you offer other explanations?

To conclude, there is still some work to be done with your manuscript. I hope that my comments would help.

Author Response

We thank the editors and reviewers of the International Journal of Environmental Research and Public Health for the thorough reviews and constructive feedback that guided the revision of our manuscript.  We hope that you agree that it has been improved by our revisions.

Below, please find point by point responses to the comments from the two reviewers.

REVIEWER 4

Introduction

Reviewer (4.1): I intended to draw your attention to the fact that you had nice data obtained with your measurement scales which allowed you to perform more advanced analysis(e.g., regression analysis). The χ2 test would show you simple associations between two nominal variables only (e.g., gender and fear, as in Table 2), while in regression analysis, you can put all your variables (including gender) together in the model, which is a more advisable approach. In my opinion, there is no reason to convert continuous (numeric) variables into nominal(dichotomized) ones; at least, I can’t see any reason provided explicitly in your manuscript. But you did so, and there is no harm in this, unless we look at the tables more carefully.

Authors (4.1): We appreciate all the reviewer's suggestions. We understand that we have carried out different ways and analyses that reinforce the main idea that associate and predict COVID-19 fear primarily by anxiety. In any case, we explain below, how we have been responding to the reviewer's concrete and rightful demands.

Reviewer (4.2): Table 2 shows a higher level of fear of the pandemic is linked in a statistically significant way to the female gender and a high degree of anxiety” [252, 253,254].

Authors (4.2): In accordance with the reviewer's suggestions, we have removed table 3 to avoid redundancy. We have partially left the comments in table 3 and we have renumbered the rest of the tables

Reviewer (4.3): The next table (Table 3) shows the results of the t-test: the mean difference between men and women as well as between low and high levels of anxiety are statistically significant [264,265], which are actually the same results as those in Table 2 (concerning two variables – gender and anxiety), but with a piece of additional information about effect size. It is easy to see some redundancy in reporting your results, which is also present in Table 5. Table 5 shows the results of the regression analysis that you performed to test two models. Each of them produces the same results; obviously, the Model II – as such as it is - is redundant. It would be more beneficial if you put in the other model an additional variable, such as gender, and apply Enter method to get more valuable information about possible predictors.

Authors (4.3):

  • Also, in accordance with the reviewer's suggestions, we have removed model II from the regression analysis to avoid redundancy. In addition, following the reviewer's good suggestions, we have included the predictor variable of gender in the linear regression analysis.
  • We have modified the comments in table 5 of the regression analysis to become table 4: “Table 4 shows the multiple linear regression analysis results applying an enter method to find the significance of the model and their predictor variables for the dependent variable "fear of COVID-19". The model is significant (p-value <0.001) and explanatory with a moderately high R (R> 0.444). The two variables that contribute the most to the model for their B coefficients and for their t statistics are anxiety (t = 9.791; p-value <0.001) and suicide risk (t = -4.172; p-value <0.001), although with a different sign, that is, greater anxiety contributes more fear of COVID-19, while the greater risk of suicide implies less fear of the pandemic. Likewise, the predictor variables, gender (woman) and social support contribute significantly and positively to the explanatory model of fear towards COVID-19. Likewise, the predictor variables gender (female) and social support contribute significantly and positively to the explanatory model of fear of COVID-19. The interaction of all the independent variables within the regression model has led to a greater relevance of low suicide risk and high social support in predicting fear of COVID-19 than was assumed in the correlation matrices.”

Table 4. Model of Multiple Linear Regression Analysis (Dependent Variable = Covid-19 fear).

B

S.E.

T

Sig. T

Constant

9.010

1.746

5.161

0.000

Gender

1.635

0.593

2.756

0.006

Anxiety

0.458

0.047

9.791

0.000

Social support

0.047

0.019

2.445

0.015

Suicide risk

-0.389

0.093

-4.172

0.000

Resilience

-0.055

0.089

-0.625

0.532

R; R2 (F; Sig.)

R=0.446; R2= 0.199 (F=25.28; p< 0.001)

Reviewer (4.4): But there are more problems with your findings. What you got with regression analysis (Table 5) is incompatible with your earlier analyses (Tables 2 & 4). I mean the finding of the role of suicide risk in the prediction of fear of COVID: In the two explanatory models, the two variables that contribute the most to the two models both for their B coefficients and for their t statistics are anxiety (t = 10.568; p-value <0.001) and suicide risk (t = -4.171; p-value <0.001), although with a different sign, that is, greater anxiety contributes more fear of COVID-19, while the greater risk of suicide implies less fear of the pandemic. How do you explain such a high association between suicide risk and fear of COVID in regression analysis, while you demonstrated no associations between these variables in former analyses (Table 2 - χ2 test & Table 4 – correlations)? Authors (4.4):

  • We share the reviewer's interest in the relationship between fear of COVID-19 and suicide risk. The possibility of the ambivalent character of "fear" as reverence or respect and terror or horror. Both produce anxiety, but respect could be a symptom of a sensible person far removed from suicide risk. Moreover, the interaction in the regression model of the predictor variable social support, which correlates positively with the fear of covid (the very likely contagious and social dimension of fear) and negatively especially with the risk of suicide (the protection of social and friendship networks against suicidal ideation), causes the risk of suicide to have a negative and significant value in the model. In any case, this hypothesis would require another research and another article.

Reviewer (4.5): In the section named “Discussion,” there is no attempt to discuss your results. You should explain your findings concerning: The positive association between fear of COVID-19 and anxiety. It would help enormously if you give clear explanations in the Introduction how you understand the concepts: fear and anxiety, and give examples of items of the employed scales (anyway, thank you for the link to your questionnaires, in Spanish though).

Authors (4.5):  Dear reviewer, following your recommendation, further information has been added in the introduction section regarding fear definition and COVID-19 fear concepts. Specifically, next information was added:

  • Hoog and colleagues [41] defined fear as an unpleasant emotional state that is triggered by the perception of threatening stimuli. Similarly, Ralph [42] considering fear as an in-tervening variable between set of context-dependent stimuli and suites of behavioral response.
  • Fear can be beneficial during the COVID-19 pandemic, or it can also be detrimental to mental and physical health. Experiencing fear can increase the risk perception, promoting the protective behaviors (washing hands and maintaining physical distance, etc.) [44]. For instances, Harper and colleagues [45] found that those individuals engage more in preventive behaviors when they perceive the threat as severe.
  • To measure fear of COVID-19, Ahorsu and colleagues [47] have developed a brief and valid instrument to capture an individual's fear of COVID-19 using a five-item Likert type scale (e.g., “I am afraid of losing my life because of coronavirus-19” and “I cannot sleep because I'm worrying about getting coronavirus-19”).

Reviewer (4.6): Although these concepts are distinct theoretically, their symptoms overlap. So, perhaps what you measure with an anxiety scale there are just symptoms of the fear of the COVID? Is it possible that these instruments measure the same phenomenon? Whatever the truth is, you should discuss possible explanations; reference to studies by other scholars is not enough.

Authors (4.6):  We totally agree with your recommendation. In this senses, further information has been added in the discussion section, specifically: “Despite of COVID-19 fear and anxiety concepts are distinct theoretically and measured in the current research using different scales their symptoms might overlap, being similar psychological reactions to COVID-19 epidemic [75]. In order to measure both phenomena in the same instrument, recently Lee and colleagues [76] have created and validated The Coronavirus Anxiety Scale, a mental health screener designed to aid researchers in identifying probable cases of dysfunctional anxiety associated with the COVID-19 crisis. Taken together, these findings urge a more in-depth exploration into the association be-tween anxiety and COVID-19 fear for future research among college population”.

Reviewer (4.7): Negative and significant correlation between suicide risk and fear of COVID (if there is a correlation!)

Authors (4.7): Following your useful suggestion, further information were added in the discussion section regarding COVID-19 fear and suicide risk: “Notably, COVID-19 fear reduced suicide risk among our sample, explained in part be-cause the possibility of the ambivalent character of "fear" as reverence or respect and terror or horror [81,82]. Both produce anxiety, but respect could be a symptom of a sensible per-son far removed from suicide risk. Moreover, the interaction in the regression model analysis of the predictor variable social support, which correlates positively with the fear of COVID-19 (the very likely contagious and social dimension of fear) and negatively especially with the risk of suicide (the protection of social and friendship networks against suicidal ideation), causes the risk of suicide might have a negative and significant value in the model. In any case, this hypothesis would require future studies”.

Reviewer (4.8):  Lack of expected associations between social support & resilience and fear of COVID

Authors (4.8): Following your useful suggestion, further information were added in the discussion section regarding social support, resilience and COVID-19 fear: “In contrast with prior studies [77] showing that higher levels of social support statically decrease COVID-19 fear, social support was positively associated with COVID-19 fear among our sample. This finding might be explained in part because fear can be contagious, thus greater social support can imply more information channels to perceive the fear of COVID-19 [78]. Contrary, COVID-19 fear had a significant inverse correlation with resilience among college students in the Philippines [79], not being statistically significant among the current sample of students in Spain. In addition, we found that greater social support and resilience reduces anxiety levels during the pandemic. In this way, and given that some of the symptoms of COVID-19 and anxiety are similar, it is recommended to deepen the studies between the analysis of these variables independently, given the bene-fits of social support and resilience on mental health found by previous authors during the COVID-19 pandemic of different age groups [80].”

Authors (4.9): Higher level of fear of COVID in women.  My guess is that women contact people suffering from COVID more often than men do, so perhaps for them, COVID is a much more real thing. Is it possible? Or can you offer other explanations?

Authors (4.9): Similarly to previous comments, further information has been added in the discussion section regarding the higher vulnerability to COVID-19 fear among women compared to men:

 “Finally, regarding demographic variables, compared with men, COVID-19 fear was statically higher among women. This finding is consistent with studies conducted among general population in Cuba [44], and also consistent with previous scholars that have re-ported greater psychological vulnerability among women compared to men during the COVID-19 pandemic [15,83]. Those women experience more fear than men during COVID-19 may be related that woman contact people suffering from COVID more often than men do, so for them COVID-19 is a much more real thing. Similarly, Alon and col-leagues [84] argued that closures of schools and day care centers have massively increased child care needs, which has a particularly large impact on working mothers and sisters among college samples.”.
